# The Effect of Epimedium Isopentenyl Flavonoids on the Broiler Gut Health Using Microbiomic and Metabolomic Analyses

**DOI:** 10.3390/ijms24087646

**Published:** 2023-04-21

**Authors:** Jiaqi Zhang, Qingyu Zhao, Yuchang Qin, Wei Si, Huiyan Zhang, Junmin Zhang

**Affiliations:** 1State Key Laboratory of Animal Nutrition, Institute of Animal Sciences, Chinese Academy of Agricultural Sciences, Beijing 100193, China; 2Scientific Observing and Experiment Station of Animal Genetic Resources and Nutrition in North China of Ministry of Agriculture and Rural Affairs, Institute of Animal Sciences of Chinese Academy of Agricultural Sciences, Beijing 100193, China

**Keywords:** epimedium, isopentenyl flavonols, gut health, immune, nutrient digestibility, serum metabolites, cecal microbiota

## Abstract

Epimedium (EM), also known as barrenwort, is a traditional medicinal plant rich in isopentenyl flavonols, which have beneficial biological activities and can improve human and animal health, but its mechanism is still unclear. In this study, ultra-high-performance liquid chromatography/quadrupole-time-of-flight-mass spectrometry (UHPLC-Q-TOF/MS) and ultra-high-performance liquid chromatography triple-quadrupole mass spectrometry (UHPLC-QqQ-MS/MS) were used to analyse the main components of EM, and isopentenyl flavonols such as Epimedin A, B, and C as well as Icariin were the major components of EM. Meanwhile, broilers were selected as model animals to illuminate the mechanism of Epimedium isopentenyl flavonols (EMIE) on gut health. The results showed that supplementation with 200 mg/kg EM improved the immune response, increased cecum short-chain fatty acids (SCFAs) and lactate concentrations, and improved nutrient digestibility in broilers. In addition, 16S rRNA sequencing showed that EMIE altered the composition of cecal microbiome, increasing the relative abundance of beneficial bacteria (*Candidatus Soleaferrea* and *Lachbospiraceae NC2004 group* and *Butyricioccus*) and reducing that of harmful bacteria (*UBA1819*, *Negativibacillus*, and *Eisenbergiella*). Metabolomic analysis identified 48 differential metabolites, of which Erosnin and Tyrosyl-Tryptophan were identified as core biomarkers. Erosnin and tyrosyl-tryptophan are potential biomarkers to evaluate the effects of EMIE. This shows that EMIE may regulate the cecum microbiota through *Butyricicoccus*, with changes in the relative abundance of the genera *Eisenbergiella* and *Un. Peptostreptococcaceae* affecting the serum metabolite levels of the host. EMIE is an excellent health product, and dietary isopentenyl flavonols, as bioactive components, can improve health by altering the microbiota structure and the plasma metabolite profiles. This study provides the scientific basis for the future application of EM in diets.

## 1. Introduction

Flavonoids are a diverse group of polyphenolic compounds derived from a flavan (2-phenylchroman) core [1]. Flavonoid polyphenols have a variety of biological functions, including antimicrobial [2], antioxidant [3], immune stimulation [4], and health improvement [5]. Flavonoids mostly occur naturally as derivatives, including glycosyl, isoprenyl, acetyl, methyl, and as polymers [6,7]. Their different structural characteristics and physicochemical properties result in differences in the type and strength of biological activity and in vivo metabolism [7,8]. Isopentenyl flavonoids are a group of flavonoids that have an isopentenyl group attached to their C-ring. Isoprenyl derivatives are particularly valuable, since their isopentenyl group increases the lipophilicity of the flavonoids and often enhances the biological activity [9]. Increased lipophilicity improves binding to the phospholipids in bacterial membranes, which exerts a strong bactericidal effect and is less likely to stimulate resistance than conventional antibiotics [10]. In addition, the low bioavailability and intestinal absorption of isopentenyl flavonols results in most of the dietary intake remaining in the digestive tract, where they can effectively increase the proportion of beneficial microorganisms and inhibit the growth of pathogenic bacteria [11,12]. 

Recent studies have suggested that isopentenyl flavonoids may also modulate the host gut microbiota and plasma metabolism. For example, icariin, an isopentenyl flavonoid from Epimedium species, has been shown to increase the abundance of beneficial bacteria (such as *Bifidobacterium* and *Lactobacillus*) and decrease the abundance of harmful bacteria (such as *Clostridium* and *Enterococcus*) in the gut of diabetic mice. Icariin also improved glucose tolerance and insulin sensitivity by regulating the expression of genes involved in glucose and lipid metabolism in the liver [13]. Similarly, sophoraflavanone G, an isopentenyl flavonoid from *Sophora flavescens*, has been reported to alter the gut microbiota composition and diversity in obese mice. Sophoraflavanone G reduced the ratio of *Firmicutes* to *Bacteroidetes* and increased the abundance of *Akkermansia muciniphila*, a mucin-degrading bacterium that has anti-obesity effects. Sophoraflavanone G also ameliorated obesity-related metabolic disorders by reducing inflammation, oxidative stress, hepatic steatosis, and insulin resistance [6]. These findings indicate that isopentenyl flavonoids may exert beneficial effects on host health by modulating the gut microbiota and plasma metabolism. However, more studies are needed to elucidate the underlying mechanisms and potential interactions between different isopentenyl flavonoids and gut microbes.

*Epimedium* L., a genus of the *Berberidaceae* family, is a well-known medicinal plant found widely in Asia, which has anti-osteoporosis [14], antibacterial [15], anti-inflammatory [16], antioxidant [17], antitumor [18], and estrogenic activities [19]. Studies have shown that Epimedium is rich in isopentenyl flavonoids and exerts biological effects [20]. In many cases, single flavonoids have limited biological activity, but combinations may produce synergistic effects [21]. In addition, the main isopentenyl flavonols of Epimedium, such as Epimedin A, B, C, and Icariin, have a very low level of toxicity, and no long-term side effects have been reported. Currently, Epimedium (EM) is permitted for use as a feed additive in China and our previous study revealed that 200 mg/kg EM significantly improved animal health [22]. However, the mechanisms by which they work are not yet clear. The aim of this study was to investigate the main isopentenyl flavonols in EM and whether their synergistic interactions affect nutrient digestibility, the immune response, the cecum microbiome, microbial short-chain fatty acids (SCFAs) production, lactic acid, and serum metabolites in broiler chickens, which may provide a reference for the use of EM as a human health product.

## 2. Results

### 2.1. Isopentenyl Flavonols Composition of EM 

The isopentenyl flavonols composition was determined by ultra-high-performance liquid chromatography triple-quadrupole mass spectrometry (UHPLC-QqQ-MS/MS) analysis; the base peak ion (BPC) chromatogram of EM in positive and negative ion modes is shown in Appendix A. Seventeen flavonoids were identified among the 30 major components of EM (Table 1).

The major EM isopentenyl flavonols were then quantified by comparison with authentic standards (Table 2). Epimedin A (39.79 g·kg^−1^), Epimedin B (91.90 g·kg^−1^), Epimedin C (110.42 g·kg^−1^), and Icariin (249.31 g·kg^−1^) were the most abundant components identified in the present study, as well as Sagittatoside A (5.758 g·kg^−1^), Baohuoside Ⅰ (14.17 g·kg^−1^), and Icartin (8.86 g·kg^−1^). These isopentenyl flavonols share the same 4′-OMe, 8-isoprenyl kaempferol backbone and differ in their glycosylation patterns (Figure 1); together, they comprised > 50% of the EM extract. 

### 2.2. Apparent Digestibilities of Nutrients 

The effects of dietary EM on apparent digestibilities of nutrients by broilers were determined (Figure 2). Compared with the NC group, the EM treatment increased crude protein and total energy absorption (*p* < 0.05), and it slightly increased dry matter absorption (*p* > 0.05). The absorption of all three nutrients was in the order EM > CTC (antibiotic treatment) > NC (untreated control), during both sampling periods.

### 2.3. Serum Immune Response 

The effects of EM on the serum immune response were determined (Figure 3). The CTC and EM groups had higher serum IgA and IgG on days 21 and 42, compared with NC (*p* < 0.05) (Figure 3A,C). Serum IL-6 and TNF-α in the EM group at days 21 and 42 were lower than the NC group (*p* < 0.05), whereas serum IL-10 in the EM group was higher than the NC group (*p* < 0.001); however, there were no significant differences in serum IL-1β (Figure 3B,D).

### 2.4. Cecal Lactic Acid and SCFAs Concentrations 

The concentrations of lactic acid, total SCFAs, and acetic and valeric acids in the cecal digesta of the EM group were higher than those in the NC and CTC groups (*p* < 0.01, *p* < 0.01, *p* < 0.01, *p* < 0.05, respectively; Figure 4). The butyric acid concentration in the EM group was higher than in the NC group (*p* < 0.05). Both the NC and EM groups had higher isovaleric acid concentrations than the CTC group (*p* < 0.05).

### 2.5. Cecum Microbiota Analysis

The Shannon index of microbial α-diversity (intra-sample diversity; Figure 5A) in the cecum was in the order EM > NC > CTC (*p* > 0.05), i.e., antibiotic treatment reduced microbial diversity, whereas EM increased it. Microbial β-diversity (diversity between samples) was compared using Principal Coordinates Analysis (PCoA) and Partial Least Squares Discriminant Analysis (PLS-DA) (Figure 5B,C, respectively); PCA produced a partial separation, whereas PLS-DA widely separated the three treatment groups, suggesting that dietary supplementation with chlortetracycline and EM had significantly different effects on the composition of the gut microbiota.

For the analysis of the microbial composition, genus was chosen as the taxonomic level. *Faecalibacterium*, *Clostridia UCG-014*, *Lachnospiraceae*, *VadinBB60_group*, *Romboutsia* and *Shuttleworthia* were the main genera (>4%) in the cecal microbiota of broilers, and *Faecalibacterium*, followed by *Clostridia UCG−014*, were the dominant genera in all of the groups (Figure 5D). In addition, significant taxa in the phylotypes were identified using linear discriminant analysis effect size (LEfSe; Figure 5E). Nine genera, three families, and two orders were identified using a Linear discriminant analysis (LDA) threshold > 2. The genera *Negativibacillus* and *UBA1819* were biomarkers in the NC group. The cecal microbiota of broilers was characterized by *CHKCI001* as a biomarker in the CTC group. The genera *norank Ruminococcaceae*, *Gordonibacter*, *Camdodatis Soleaferrea*, and *Lachnospiraceae NC2004 group* were biomarkers in the EM group. The Phylogenetic Investigation of Communities by Reconstruction of Unobserved States (PICRUSt2) analysis showed that the dietary EM treatment induced higher heat map scores than NC in the known functional genes for the following genes, cell cycle control, cell division, chromosome partitioning, energy production and conversion, amino acid transport and metabolism, lipid transport and metabolism, posttranslational modification, protein turnover, chaperones, signal transduction mechanisms, and nucleotide transport and metabolism (Figure 5F). EM treatment also induced higher heatmap scores than CTC in the known functional genes for transcription, defense mechanism, replication, recombination and repair, translation, ribosomal structure and biogenesis, intracellular trafficking, secretion, and vesicular cell cycle control, cell division, chromosome partitioning, energy production and conversion, amino acid transport and metabolism, and lipid transport and metabolism.

### 2.6. Blood Serum Metabolome 

To investigate metabolic regulation in EM-treated birds, serum metabolites were analysed by UHPLC-Q Exactive HF-X. Identified metabolites were manually analysed to eliminate false positives based on m-scores, chromatographic peak shapes, and in-source fragmentation to increase the reliability of the output results. A total of 653 metabolites (396 in positive mode and 257 in negative mode) were confirmed. The five most abundant metabolites detected were carboxylic acids and derivatives (22.43%), glycerophospholipids (18.24%), fatty acyles (10.48%), and prenol lipids (5.66%) (Figure 6A).

To differentiate between the broiler metabolite profiles under the different treatments, the data were subjected to multivariate analysis. The NC and CTC groups were partially separated by PCA (Figure 6B), but well separated from the EM group, whereas the NC, CTC, and EM groups were all well separated by PLS-DA (Figure 6C), indicating that there was a significant difference in serum metabolite composition between the three groups. To identify the differential metabolites between the three groups, they were screened with limits of fold-change > 2 and *p* < 0.05 (Appendix A and Figure 7), which identified a total of 48 significant differential metabolites. Two CTC group metabolites were up-regulated (fold-change > 2) and 10 were down-regulated (fold-change < 0.5) compared with the NC group. Seven EM group metabolites were up-regulated and 41 were down-regulated compared with the NC group. The top five of the value improvement process (VIP) values were calculated for the differential metabolites to determine those that changed the most (Appendix A). Dietary EM markedly decreased serum levels of pantothenic acid, L−methionine−S-oxide, and the relative levels of 1−(1,2,3,4, 5−pentahydroxypent−1−xyl)-1,2,3,4-tetrahydro-beta-carboline 3-carboxyl, and pyrocatechol sulfate (*p* < 0.05), but they increased the relative levels of 5−(3′,5′−dihydroxyphenyl)−γ−valerolactone−O−sulfurate−O−methyl (*p* < 0.001). Kyoto encyclopedia of genes and genomes (KEGG) topology analysis of differential metabolites indicated that the EM affects broiler immunity and apparent digestibilities of nutrients, mainly through functions such as pantothenate and CoA biosynthesis, β-alanine metabolism, and cysteine and methionine metabolism (Appendix A).

### 2.7. Co-Occurrence Network

To explore the relationship between changes in the microbiota and serum metabolome, a correlation analysis was performed between the top 30 differential cecal bacterial genera and the differential serum metabolites (Appendix A). *Eisenbergiella* had the strongest association with serum metabolites; for example, *Eisenbergiella* and diallyl trisulfide were negatively correlated with 5−(3′,5′−dihydroxyphenyl)−γ−valerolactone−O−sulfurate−O−methyl (*p* < 0.05, r < −0.7). *Lactobacillus* was negatively correlated with S−adenosylhomocysteine (*p* < 0.05 and r < −0.6).

To show the relationships between bacterial genera and serum metabolites more intuitively, the correlations between them were visualized with Cytoscape. With limits of the absolute value of the correlation coefficient, r > 0.6 and *p* < 0.05, 51 correlations were obtained, including 11 correlations between different bacterial genera, 23 correlations between bacterial genera and metabolites, and 14 correlations between different metabolites (Figure 8A). *Eisenbergiella* and *Un. Peptostreptococcaceae* had the largest number of strong correlations with metabolites and were core bacterial genera in the genus relationship network. *Butyricicoccus*, erosnin, and tyrosyl-tryptophan were the core genera and core metabolites in the genus relationship network and metabolite relationship network, respectively, and appear to be important regulators of the whole network. The EM group had reduced cecal abundance of *Eisenbergiella* (*p* > 0.05) and increased relative abundance of *Un. Peptostreptococcaceae* and *Butyricicoccus* (*p* < 0.05, *p* > 0.05; Figure 8B–D), compared with NC. The EM group also had markedly reduced relative serum levels of erosnin and tyrosyl-tryptophan (*p* < 0.01; Figure 8E,F) compared with CTC and especially NC.

## 3. Discussion

Flavonoids are plant secondary metabolites and occur in a wide range of fruits and vegetables, but they share a common carbon skeleton of two phenyl rings (A and B) (Figure 1) joined by a pyran ring (C) [23]. The main classes of flavonoids, chalcones, flavones, flavonols, flavanols, flavans, flavanones, anthocyanidins, and isoflavonoids are distinguished by the substitution patterns on the heterocyclic C−ring [24]. The isopentenyl flavonols, Epimedin A, B and C, Icariin, Sagittatoside A, Baohuoside Ⅰ, and Icartin were the predominant EM components identified. All of these compounds possess an isopentenyl group at position 8, a glucose group at position 7-O, and a different sugar group at position 3−O (Figure 1) [25]. Isopentenyl derivatization increases the hydrophobicity of natural products, thereby increasing their affinity for cell membranes and improving their antimicrobial activity [26,27]. The high affinity of isopentenyl flavonols for cell membranes can dissipate the proton-motive force in mitochondrial membranes and disrupt other metabolic processes, thereby avoiding the development of antibiotic resistance [10]. The results from this study suggest that isopentenyl flavonols are a potential alternative to antibiotics for animal husbandry and may inform the use of EMIE (Epimedium isopentenyl flavonols) as a human health product. [28,29,30].

In our previous study, we found that the optimal gradient of EM to improve animal health was 200 mg/kg [21], so we selected 200 mg/kg for a deeper study. EMIE may enhance immune function through phosphatidylinositol 3-kinase (PI3K)/serine-threonine protein kinase (AKT) signaling and increase serum estradiol and IL-2 [31]. Epimedium flavonoids ameliorated hypertension in rats by reducing serum concentrations of pro-inflammatory TNF-α [32]. Supplementation of 200 mg/kg of EM in the basal broiler diet increased serum IgA and IgG levels and decreased the levels of pro-inflammatory factors IL-1β, IL-6 and TNF-α, and it increased the levels of anti-inflammatory factor IL-10, possibly by modulating the intestinal mucosal immune system and the relative abundance of intestinal micro-organisms, which, in turn, improved host immunity [33]. The mechanism by which EMIE modulates microorganisms to achieve anti-inflammatory effects remains to be elucidated.

Isoprenylation increases the hydrophobicity of flavonoids, thereby improving their bioavailability, but their bioavailability is still low and limits their potential pharmacological applications [9,34]. However, low intestinal absorption of dietary isopentenyl flavonoids results in high concentrations in the colon, which can modify the relative abundance of intestinal microbial species through their antimicrobial properties, and which facilitates microbial metabolism into other bioactive compounds. Gut microbiota α-diversity, based on operational taxonomic unit (OTU) relative abundance, was altered by dietary EM. Dietary supplementation with chlortetracycline decreased the Shannon index of cecum microorganisms, whereas EM supplementation increased it. This suggests that EM increases cecum microbial abundance, whereas tunicamycin decreases it. The cecum microbial β-diversity was compared by PCA and PLS-DA analysis, which showed that the three treatments produced distinct and clearly differentiated microbiota.

The gut microbiota is an important component of the intestinal immune system. A balance of commensal and probiotic bacteria promotes the integrity of the intestinal epithelial barrier, thereby helping to protect the host from infection by pathogenic bacteria [35,36]. In this study, *Faecalibacterium* was the predominant genus, in agreement with a previous report [37]. Dietary supplementation of broilers with 200 mg/kg of EM increased the relative abundance of *Faecalibacterium*, a highly metabolically active symbiotic bacterium that has an important function in balancing intestinal immunity [38]. Dietary supplementation with EM also increased the abundance of the *Gordonibacter*, *Candidatus Soleaferrea*, and *Lachnospiraceae NC2004 group* genera. Degradation of low-bioavailability flavonoid glycosides depends on the actions of intestinal microorganisms. Catabolic activity by *Gordonibacter,* catalyzing the opening of the flavonoid C-ring and the breaking of the C-C bond, allows the conversion of non-absorbed flavonoids into more bioavailable metabolites [39]. The genus *Candidatus Soleaferrea* has anti-inflammatory effects through the secretion of metabolites and assisting maintenance of intestinal homeostasis, which improves host immunity [40]. *Lachnospiraceae NC2004 Group* are the main anaerobic bacteria in the microbiota of healthy populations, and are capable of producing a range of beneficial metabolites, such as SCFAs [41], involved in the production of indole-3-acetic acid [42], the conversion of primary bile acids to secondary bile acids, and resistance to pathogen colonization [43]. EM was able to reduce the relative abundance of the harmful bacteria *UBA1819* and *Negativibacillus* in the cecum contents; both are pathogens responsible for many diseases, and the reduction in their abundance would help to maintain a healthy intestinal microbial balance [44,45]. Therefore, dietary EMIE supplementation and the microbiota have a mutual regulatory effect; the great variety of microbial enzymes metabolize flavonoids and increase their bioactivity, and EMIE increases the proportion of beneficial intestinal bacteria, improving the microbiota structure.

Changes in the relative abundance of intestinal microorganisms are clearly correlated with their metabolites. Flavonoids have been shown to exert a prebiotic effect by stimulating microbial production of lactic acid and short-chain fatty acids [46], which is consistent with the findings of this study. The relative abundance of *Lachnospiraceae NC2004 group* and *Butyricicoccus*, which are producers of SCFAs, increased in the EM group. The EM group had higher levels of cecal acetic, butyric, valeric and lactic acids, and total SCFAs compared with the NC group. Lactic acid is an important compound that links food and the intestinal microbiota [47] and can also be utilized by some bacteria to produce butyric acid through a cross-feeding mechanism. SCFAs, especially butyrate, have important immunomodulatory functions and contribute to the maintenance of intestinal homeostasis as well as being an excellent carbon source for intestinal epithelial cells. 

The biological functions of the cecum microbiota were predicted by PICRUSt2, revealing that those of the EM group were more focused on amino acid transport and metabolism, lipid transport and metabolism, energy production and conversion, cell cycle control, cell division, and chromosome partitioning. This suggests that EM modulation of the intestinal microbiota results in the microbiome organisms helping improve the digestion and the apparent digestibility of nutrients as well as the immunity of the host. Therefore, 200 mg/kg EM selectively promotes the growth and proliferation of beneficial intestinal bacteria and increases the content of their bioactive metabolites, thereby enhancing intestinal digestion and the apparent digestibility of nutrients.

The very diverse intestinal microbiota has an extensive metabolic capacity that complements the activity of animal enzymes in the liver and intestinal mucosa [48], and that makes an important contribution to host metabolism by contributing enzymes not encoded by the host genome (e.g., for synthesis of polysaccharides, polyphenols, and vitamins) [49], which is consistent with the results from PICRUSt2 analysis. There is growing evidence that flavonoids affect glucose, protein, amino acid, and lipid metabolism, which is also somewhat consistent with the findings of this study. Dietary supplementation with EM affected broiler metabolism mainly through improvements in β-alanine metabolism, pantothenate, CoA biosynthesis, and cysteine/methionine metabolic pathways. In addition, the differential metabolites were correlated with the 30 most abundant genera. According to the principle of mediocentricity, the genus *Butyricoccus* was the core genus, Erosnin and tyrosyl-tryptophan were the core serum metabolites in the relationship network, and the genera *Eisenbergiella* and *Un. Peptostreptococcaceae* were associated with strong correlations between metabolites. Thus, EM modulates the cecal microbiota mainly by increasing the relative abundance of *Butyricicoccus*, which, in turn, affects the host serum metabolite profile by altering the relative abundance of *Eisenbergiella* and *Un. Peptostreptococcaceae*. Erosnin and tyrosyl-tryptophan are, therefore, potential biomarkers to evaluate the effects of EMIE and provide a scientific basis for the future application of EMIE.

## 4. Materials and Methods

### 4.1. Chemicals and Reagents

Analytical standards of Epimedin A, Epimedin B, Epimedin C, Icariin, Icartin, Baohoupside Ⅰ, Sagittatoside A, Sagittatoside B, and 2′-O-rhamnosyl icariside II were from Shanghai Yuanye Bio-Technology (Shanghai, China), of a purity of >98%. Methanol, formic acid, and acetonitrile were provided by Merck KGaA (Darmstadt, Germany). Ultrapure water was used for analysis. Excellent grade pure HNO_3_ and H_2_SO_4_ (National Pharmaceutical Group Chemical Reagent Co., Ltd., Beijing, China), BV-III grade H_2_O_2_ (Beijing Institute of Chemical Reagents, China Beijing), and the EM plant material were provided by Jinpai (Huangshi, China). 

### 4.2. Epimedium Extracts for Dietary Supplementation and Analysis

For analysis, EM (0.1 g) was extracted with aqueous methanol (40 mL, 70% *v*/*v*) at 35 °C in an ultrasonic bath for 30 min, and then followed by centrifugation at 10,000× *g* at 4 °C for 30 min. For qualitative and quantitative analysis, the supernatant was filtered through a 0.22 μm cellulose membrane (Merck). For the qualitative analysis of the main components of EM, chromatography/quadrupole-time-of-flight-mass spectrometry (UHPLC-Q-TOF/MS) was used. The flavonoid profile analysis was performed using ultra-high-performance liquid chromatography triple-quadrupole mass spectrometry (UHPLC-QqQ-MS/MS). Full details of the methods are reported in Appendix A and Appendix A.

Peak matching was performed using Agilent MassHunter Qualitative Analysis software (version 10.0, Agilent, Santa Clara, CA, USA), and the chemical components of EM were identified from molecular masses, and retention times and fragmentation were analyzed by Sirius (version 5.6.3, Potsdam, Germany) and MassHunter Molecular Structure Correlation software (version 8.2, Agilent, Santa Clara, CA, USA) and compared with MS data from the literature.

### 4.3. Birds and Experimental Design

Birds and the experimental design were as described previously [22]. In total, 360 (AA+) 1-day-old chickens were randomly put into three groups (8 replicates, 15 birds per replicates kept in the same pen). During the 42-day test period, the basal diet was given to the normal control (NC) group; the basal diet was supplemented with 75 mg/kg chlortetracycline and was fed to the antibiotic (CTC) group; and the basal diet supplemented with 200 mg/kg EM extract was fed to the EM group. All of the birds were housed in suspended pens with free access to food and water and 16 h of light per day throughout the 42-day experiment. The experimental diets were formulated (Appendix A) to meet the nutritional requirements of broiler chicks in accordance with the recommendations of the National Research Council (1994) [50] and the reference tables of feed ingredients for feed composition and nutritional values in China [51]. All experimental protocols were approved by the Animal Care and Use Committee, Institute of Animal Science, and the Chinese Academy of Agricultural Sciences (IAS2021-71), and trials were conducted according to relevant guidelines and regulations.

### 4.4. Sample Collection

At 21 and 42 days of age, two chickens were randomly selected from each cage and 4 mL of blood was collected from the left wing vein. Blood was left at room temperature for 30 min before centrifugation at 3000× *g* for 30 min. At 42 days of age, the chickens were killed after blood collection and both cecum were re-moved immediately. Cecum contents were collected for 16S rRNA gene amplicon sequencing (microbiota) analysis. All samples were stored at −80 °C.

### 4.5. Apparent Digestibilities of Nutrients

Cr_2_O_3_ (Chromium oxide, Cr_2_O_3_) (0.3%, as a digesta marker) was evenly mixed into the diets during 14–21 d and 35–42 d, and chicken manure was collected for 3 consecutive days from the 18th and 39th days of the experiment. After sulfuric acid treatment, it was dried at 65 °C, placed at room temperature for 24 h, and pulverized; the Cr_2_O_3_ content, moisture, total energy, and crude protein, etc. in the feed and droppings were measured, and the percentage absorption of each nutrient was calculated.

The concentration of chromium (Cr) in the sample was determined following the Chinese national standard method (GB5009.268-2016) [52]. Briefly, about 0.2 g of the sample was weighed and added into a digestion tube that contained 8 mL of nitric acid and 2 mL of hydrogen peroxide. The mixture was digested for 4 h to 6 h at room temperature using a microwave digestion instrument (CEM-MARSX^®^, CEM Corporation, Matthews, NC, USA). The digested solution was then diluted with ultrapure water to a desired concentration and filtered through a 0.45 μm filter. The Cr content in the solution was measured using an Agilent 7900 ICP-MS system (Agilent Technologies, Santa Clara, CA, USA) and converted to Cr_2_O_3_.

The crude protein content of the samples was determined using the Kjeldahl method according to the Chinese national standard method (GB/T 6432-2018) [53]. Meanwhile, the total energy content of the samples was determined using oxygen bomb combustion calorimetry following the Chinese group standard (T/NAIA 0006-2020) [54].

### 4.6. Immune Markers

The serum and mucosal cytokines, including interleukin-6 (IL-6), IL-10, IL-1β and tumor necrosis factor-α (TNF-α), as well as the serum immunoglobulins (immunoglobulin A (IgA) and IgG), were measured using the specific Enzyme-Linked Immunosorbent Assay (ELISA) kits (ML Bio, Shanghai, China) according to the manufacturer’s instructions.

### 4.7. Cecal Lactic Acid and SCFAs Analysis

A droppings sample of approximately 0.5 g was diluted with ultrapure water (2 mL), centrifuged (14,000× *g*, 4 °C for 10 min), and filtered through a 0.22 μm microporous membrane. Lactic acid content was determined using an Agilent 6470 HPLC-MS (Agilent Technologies, Santa Clara, CA, USA). Full details of the methods are reported in Appendix A.

A droppings sample of approximately 0.5 g was diluted with ultrapure water (2 mL) and centrifuged (10,000× *g*, 15 min at 4 °C). A supernatant (900 μL) was mixed with ice-cold metaphosphoric acid solution (100 μL, 25% *w*/*v*) solution at 4 °C for 4 h, centrifuged (10,000× *g*, 15 min), and then the supernatant was filtered through a 0.45-μm microporous membrane. The SCFAs content (acetic, propionic, isobutyric, butyric, isovaletic, and valeric acids) was determined by gas chromatography with DB-FFAP column (30 m × 250 μm × 0.25 μm, Agilent). The carrier gas was N_2_ (12.5 MPa, 0.8 mL/min). The flame ionisation detector temperature was 280 °C and the column temperature was increased from 60 °C to 220 °C at 20 °C/min.

### 4.8. The 16S rRNA High-Throughput Sequencing

Cecum contents were collected for genomic Deoxyribonucleic acid (DNA) extraction on day 42 of the experiment using a QIAamp DNA droppings Mini Kit (Qiagen, Hilden, Germany) according to the manufacturer’s instructions. DNA was amplified by a Polymerase Chain Reaction (PCR) using Q5 High-Fidelity DNA Polymerase (NEB, Ipswich, MA, USA) with V_3_-V_4_ region primers (338F: 5′-ACTCCTACGGGAGGCAGCAG-3′; 806R: 5′-GGACTACHVGGGTWTCTAAT-3′). The Illumina MiSeq platform (Shanghai Majorbio Bio-pharm Technology Co., Ltd., Shanghai, China) and MiSeq Reagent Kit version 3 were used to purify, quantify, and sequence the PCR products. Microbiota analysis was performed using QIIME2 (version 2021.8, http://qiime2.org) and R software (version 3.6.3, https://www.r-project.org/, Accessed on 17 October 2022). Raw reads were deposited in the NCBI Sequence Read Archive database (https://dataview.ncbi.nlm.nih.gov/object/PRJNA904521, Accession Number: PRJNA904521, Accessed on 28 November 2022). The 16S functional prediction analysis was performed using PICRUSt2 (version 2.2.0) software. Full details of the methods are reported in Appendix A.

### 4.9. UHPLC-Q Exactive HF-X Analysis of Serum Metabolites

Plasma samples from 24 broilers (6 from each treatment group) were subjected to metabolomic analysis with a UHPLC-Q Exactive HF-X system (Thermo Fisher Scientific, Waltham, MA, USA) to examine changes in non-targeted metabolites. Full details of the methods are reported in Appendix A.

### 4.10. Co-Occurrence Network Analysis 

The co-occurrence patterns of the cecum microbiota (relative abundance) and serum metabolites (relative peaks intensities) were analyzed using network analysis. Pairwise, Spearman’s rank was used to construct a correlation matrix. Relationships between metabolites were considered significant with an absolute Spearman’s correlation coefficient >0.9, (or <−0.9) and *p* < 0.05, and the relationship between bacteria and metabolites met the requirements of the correlations with an absolute Spearman’s correlation coefficient >0.6, (or <−0.6) and *p* < 0.05. The calculated *p*-values less than 0.05 were transformed into associations between the microbiota in the cecum and the metabolites in the serum in co-occurrence networks. Cytoscape (version 3.9.1, https://cytoscape.org/, Accessed on 3 November 2022) was then used to visualise the co-occurrence networks. 

### 4.11. Statistical Analysis

Data were analysed by one-way ANOVA using SAS 9.4 software (SAS Institute, Cary, NC, USA), and *p* < 0.05 was considered statistically significant. All data were expressed as mean and pooled SEM. The R package pheatmap was used for heat map analysis, and GraphPad Prism 8.0 (GraphPad Software, San Diego, CA, USA) was used to generate the histogram. Apparent nutrient digestibility (%) was calculated according to the following formula [55]:(1)$$\mathrm{Apparent}\;\mathrm{nutrient}\;\mathrm{digestibility}\;\left(\%\right)=100\%-100\%\times ({\left(Ti\right)}_{feed}/{\left(Ti\right)}_{feces})\times ({\left(nutrient\right)}_{feces}/{\left(nutrient\right)}_{feed})$$

## 5. Conclusions

EM is an excellent dietary source of isopentenyl flavonols. Dietary supplementation of the basal broiler diet with 200 mg/kg EM extract had beneficial effects on broiler health, including increasing the apparent digestibility of nutrients, enhanced immunity, increasing the abundance of the beneficial bacteria *Butyricoccus*, *Lachbospiraceae NC2004 Group* and *Candidatus Soleaferrea* spp. in the cecum, and increasing cecal SCFA and lactic acid content. EM extract appears to regulate the cecal microbiota through *Butyricicoccus*, which, in turn, influences host metabolism and changes the relative serum levels of the potential biomarkers Erosnin and tyrosyl-tryptophan, ultimately improving broiler health.

## Figures and Tables

**Figure 1 ijms-24-07646-f001:**
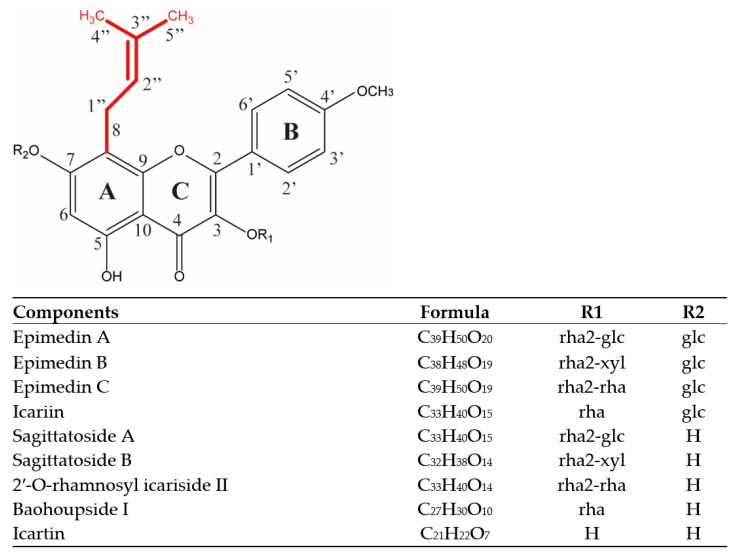
Structures of main isopentenyl flavonols found in Epimedium.

**Figure 2 ijms-24-07646-f002:**
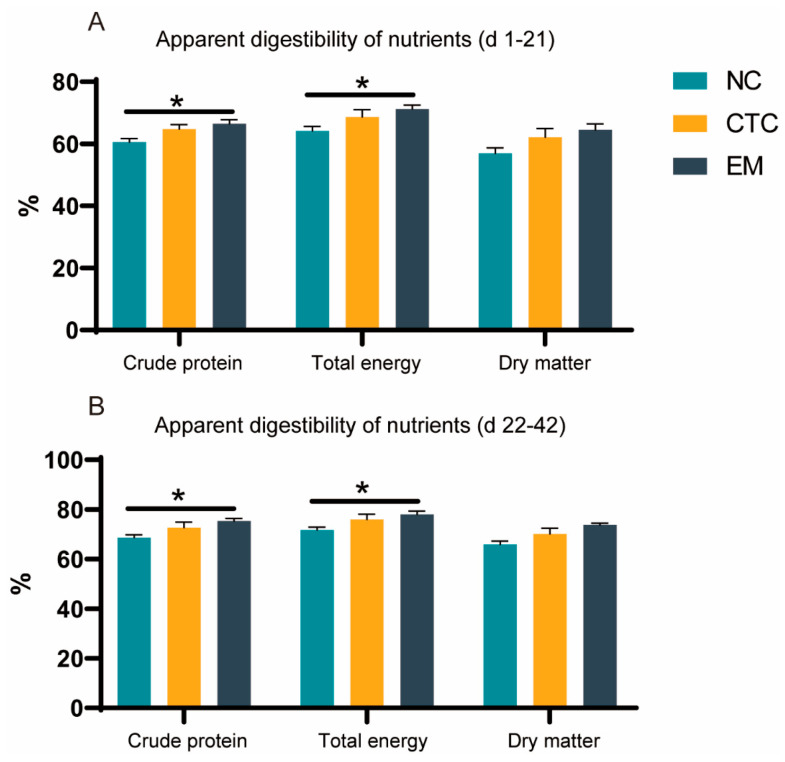
Effects of CTC and EM on the apparent intestinal absorption of crude protein, total energy, and dry matter in broilers during days 1–21 (**A**) and 22–42 d of the trial (**B**). * *p* < 0.05, compared with control (NC). N = 6 per treatment group.

**Figure 3 ijms-24-07646-f003:**
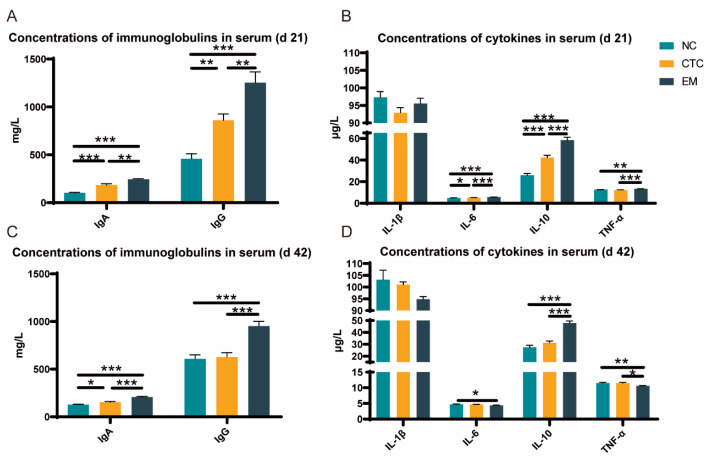
Effect of NC, CTC, and EM on serum immunoglobulins (**A**) and serum cytokines (**B**) at 21 d, and serum immunoglobulins (**C**) and serum cytokines (**D**) at 42 d in broilers. Significant differences were regarded as * *p* < 0.05, ** *p* < 0.01, *** *p* < 0.001. N = 6 per treatment group.

**Figure 4 ijms-24-07646-f004:**
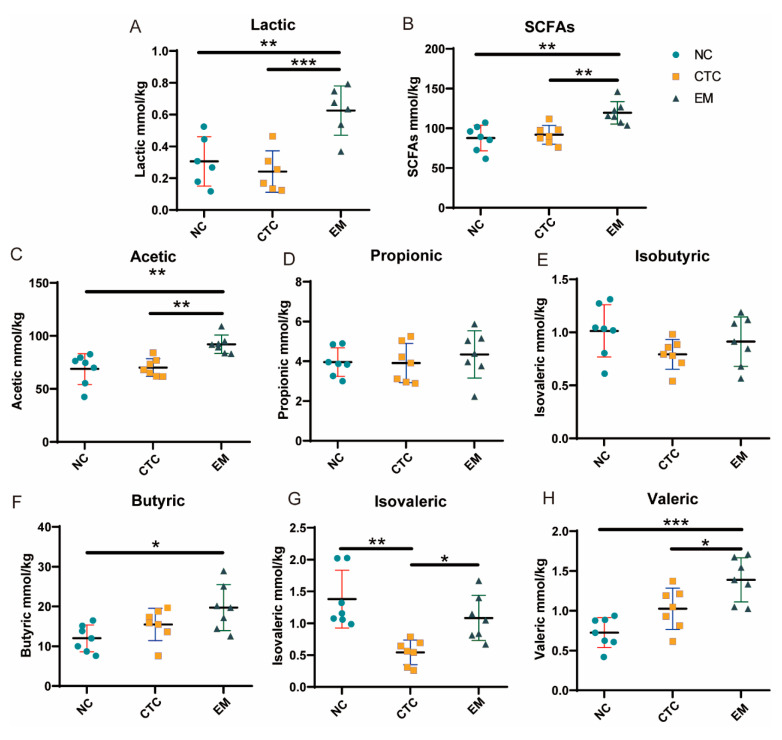
Effects of NC, CTC and EM on the concentrations of (**A**) lactic acid, (**B**) total SCFAs, (**C**) acetic acid, (**D**) propionic acid, (**E**) isobutyric acid, (**F**) butyric acid, (**G**) valeric acid, and (**H**) isovaleric acid in broiler cecum. * *p* < 0.05, ** *p* < 0.01, *** *p* < 0.001. N = 6 per treatment group.

**Figure 5 ijms-24-07646-f005:**
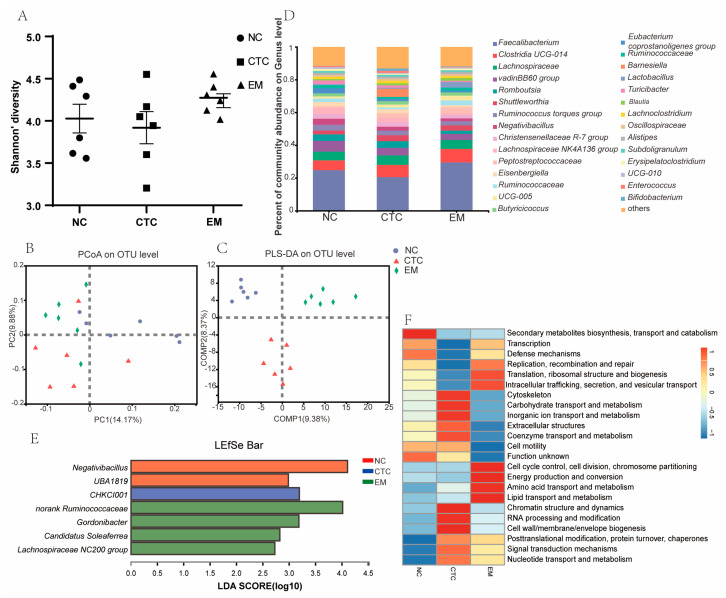
Summary of microbiota composition in the cecal contents of broilers on day 42. (**A**) Shannon Index reflecting species α-diversity within groups. (**B**) Component microbial β-diversity compared by PCA. (**C**) PLS-DA-based comparison of samples between multiple groups and (**D**) the genus-level microbiota composition. (**E**) LDA scores generated for differentially enriched genera with limits of LDA > 2, *p* < 0.05. (**F**) PICRUSt2 analysis. N = 6 per treatment group.

**Figure 6 ijms-24-07646-f006:**
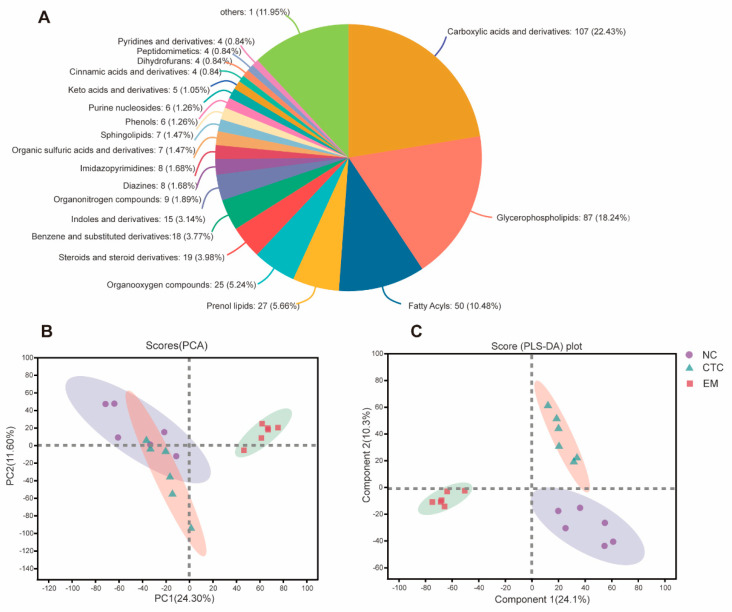
(**A**) Sector diagram showing the classification of broiler serum metabolites based on the HMDB 4.0 database. (**B**) PCA plots of broiler serum metabolomes. (**C**) PLS-DA plot of broiler serum metabolomes.

**Figure 7 ijms-24-07646-f007:**
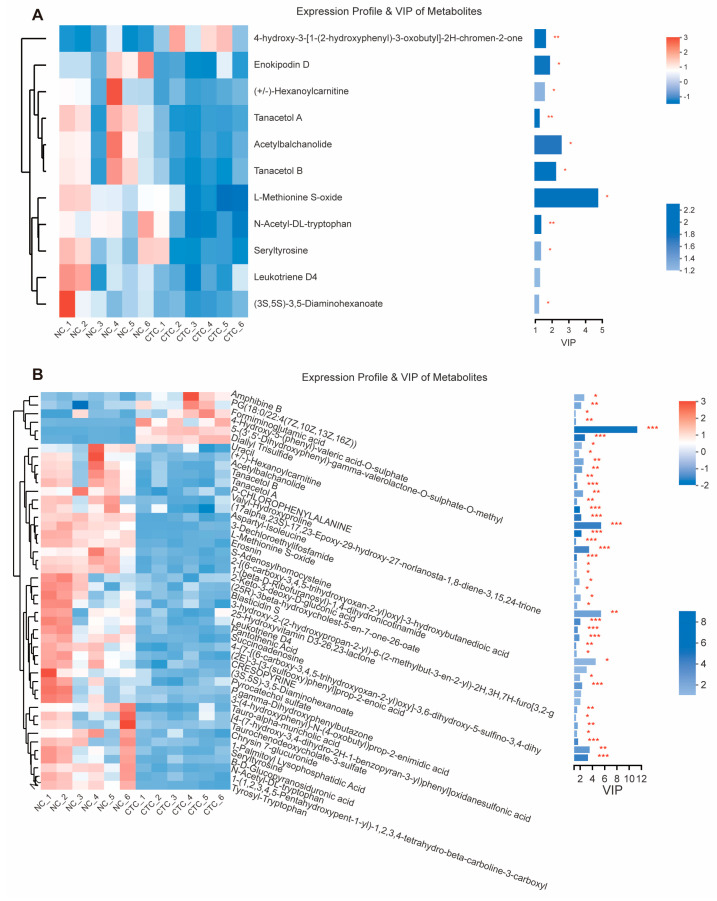
Panels (**A**,**B**) show the relative abundance of differential metabolites and their VIP values for dietary supplementation of CTC and EM, compared with the NC group. * *p* < 0.05, ** *p* < 0.01, *** *p* < 0.001. N = 6 per treatment group.

**Figure 8 ijms-24-07646-f008:**
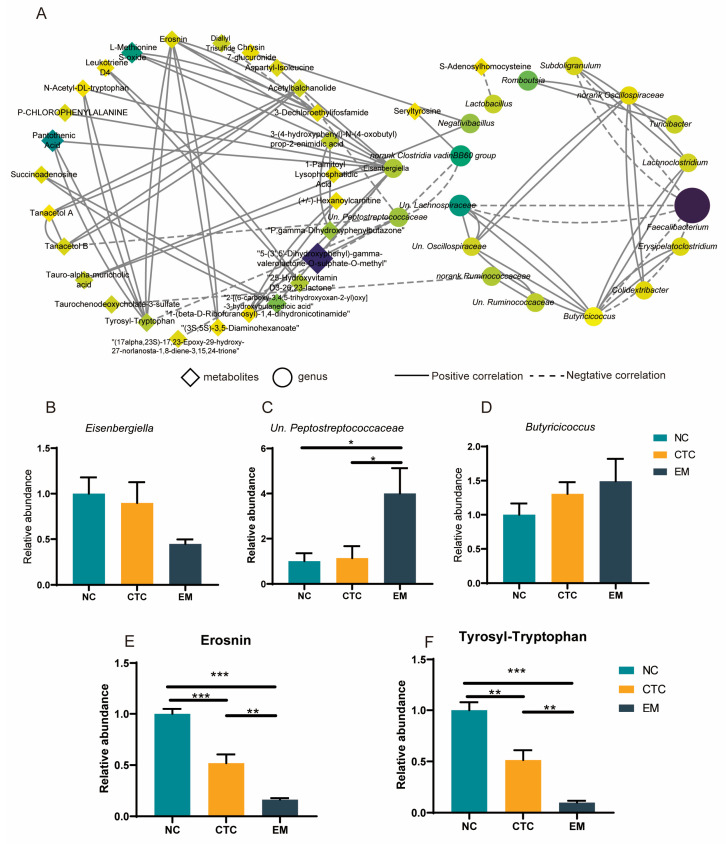
(**A**) Network analysis of the 30 most abundant cecal microbial genera with differential serum metabolites in the NC, CTC, and EM groups. Relative abundance of core genera *Eisenbergiella* (**B**), *Un. Peptostreptococcaceae* (**C**) and *Butyricoccus* (**D**). Relative abundance of the core serum metabolites Erosnin (**E**) and tyrosyl−tryptophan (**F**) among the treatment groups. Diamonds represent the differential serum metabolites and the circles represent the 30 most abundant cecal microbial genera. Solid lines represent positive correlations, and dashed lines represent negative correlations. Significant differences were regarded as * 0.01 < *p* ≤ 0.05, ** 0.001 < *p* ≤ 0.01, *** *p* ≤ 0.001. N = 6 per treatment group.

**Table 1 ijms-24-07646-t001:** Flavonoid compounds tentatively identified in EM extract, from UHPLC-Q-TOF/MS.

No.	Identity	Formula	RT (min)	MS (*m*/*z*)	Adduct Ions	Main MS/MS Fragments Detected (Arranged from Large to Small According to Relative Intensity)
1	Quercetin 3-neohesperidoside	C_27_H_30_O_16_	4.70	610.1548	[M−H]^−^	300.0275[M-2H-C_12_H_20_O_9_]^−^; 301.0350[M-H-C_12_H_20_O_9_]^−^
2	Lespedin	C_27_H_30_O_14_	4.91	578.1644	[M−H]^−^	287.0547[M-H-C_9_H_18_O_8_]^−^; 433.1117[M-H-C_6_H_8_O_4_]^−^
3	Rouhuoside	C_38_H_48_O_20_	5.41	824.2753	[M−H]^−^	661.2147[M-H-C_6_H_10_O_5_]^−^
4	Epimedoside D	C_37_H_46_O_19_	5.52	794.2644	[M−H]^−^	631.2038[M-H-C_6_H_10_O_5_]^−^
5	Ikarisoside B	C_32_H_38_O_15_	5.68	662.2221	[M−H]^−^	353.1030[M-H-C_12_H_22_O_9_]^−^; 351.0874[M-3H-C_12_H_26_O_9_]^−^; 514.1484[M-2H-C_6_H_14_O_4_]^−^
6	Epimedoside A	C_32_H_38_O_15_	5.73	662.2200	[M−H]^−^	355.1171[M+H-C_12_H_20_O_9_]^+^; 517.1693[M+H-C_6_H_10_O_4_]^+^
7	Epimedin A	C_38_H_48_O_20_	6.31	824.2756	[M−H]^−^	675.2301[M-H-C_5_H_8_O_5_]^−^
8	Epimedin C	C_39_H_50_O_19_	6.44	822.2856	[M+H]^+^	531.1854[M+H-C_12_H_20_O_8_]^+^; 677.24327[M+H-C_6_H_10_O_4_]^+^; 369.1326[M+H-C_18_H_30_O_13_]^+^
9	Epimedin B	C_38_H_48_O_19_	6.45	809.2856	[M+H]^+^	369.1324[M+H-C_17_H_28_O_13_]^+^; 531.1847[M+H-C_11_H_18_O_8_]^+^; 677.2426[M+H-C_5_H_8_O_4_]^+^
10	Icariin	C_33_H_40_O_15_	6.76	677.2400	[M−H]^−^	367.1190[M-H-C_12_H_20_O_9_]^−^; 531.1774[M-H-C_6_H_10_O_5_]^−^; 529.1724[M-H-C_6_H_10_O_4_]^−^
11	Sagittatoside A	C_33_H_40_O_15_	6.97	677.2400	[M−H]^−^	367.1186[M-H-C_12_H_20_O_9_]^−^; 513.1774[M-H-C_6_H_10_O_5_]^−^; 529.1718[M-H-C_6_H_10_O_4_]^−^
12	Baohuoside II	C_26_H_28_O_10_	8.13	500.1695	[M+H]^+^	352.0954[M-2H-C_6_H_10_O_4_]^−^; 353.1029[M-H-C_6_H_10_O_4_]^−^
13	Sagittatoside B	C_32_H_38_O_14_	8.86	646.2255	[M+H]^+^	369.1328[M+2H-C_6_H_6_O_4_]^+^
14	2″-O-Rhamnosyl ikarisoside A	C_32_H_38_O_14_	8.87	646.2271	[M−H]^−^	366.1106[M-2H-C_12_H_18_O_8_]^−^; 351.0872[M-3H-C_12_H_20_O_8_]^−^; 367.1180[M-H-C_11_H_18_O_8_]^−^
15	Anhydroicaritin 3-Rhamnosyl-(1->2)-Rhamnoside	C_33_H_40_O_14_	8.95	660.2426	[M−H]^−^	366.1108[M-2H-C_12_H_20_O_8_]^−^; 351.0871[M-3H-C_13_H_22_O_8_]^−^; 367.1182[M-H-C_12_H_20_O_8_]^−^
16	Baohuoside I	C_27_H_30_O_10_	9.45	515.1906	[M+H]^+^	369.1327[M+H-C_6_H_9_O_5_]^+^; 313.0702[M+H-C_6_H_9_O_5_]^+^
17	Icartin	C_21_H_20_O_6_	9.46	369.1327	[M+H]^+^	313.0702[M+H-C_4_H_6_O6]^+^

**Table 2 ijms-24-07646-t002:** List of main isopentenyl flavonols and their MRM parameters, from UHPLC-QqQ-MS/MS.

Target Compounds	Ion Mode	RT (min)	Precursor Ion (*m*/*z*)	Product Ions (*m*/*z*)	DP (V)	CE (V)	Quantification Transition	Compound Concentration (g/kg)
Epimedin A	+	4.609	839.3	369.1	150	45	839.3→369.1	39.7908
				313.1	150	77		
Epimedin B	+	5.211	809.3	369.1	150	41	809.3→369.1	91.9008
				313.1	150	74		
Epimedin C	+	5.882	823.3	369.1	150	44	823.3→369.1	110.4184
				313.1	150	76		
Sagittatoside A	+	12.761	677.7	369.5	110	8	677.7→369.1	5.758
				313.3	110	46		
Sagittatoside B	+	12.963	646.5	369.1	120	40	646.5→369.1	ND
				325.3	120	48		
2′-O-rhamnosyl icariside II	+	12.999	661.3	369.4	90	20	661.3→369. 1	ND
				369.1	90	20		
Icariin	+	6.727	677.2	313.3	110	46	677.2→313.1	249.3108
				313.1	120	62		
Baohuoside Ⅰ	+	12.428	515.2	369.1	150	17	515.2→369.1	14.1668
				313.1	150	45		
Icartin	+	13.793	387.5	369.3	120	22	387.5→369.1	8.8632
				313.4	120	30		

RT, retention time; DP, declustering potential; CE, collision energy.

## Data Availability

All data in both manuscript and Appendix A.

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
