# Peer review of "The Effect of Epimedium Isopentenyl Flavonoids on the Broiler Gut Health Using Microbiomic and Metabolomic Analyses"

_ijms, 2023, doi:10.3390/ijms24087646_

Round 1

Reviewer 1 Report

The present study was conducted to investigate the main isopentenyl flavonols in EM and whether their synergistic interactions affect nutrient digestibility, the immune response, the cecum microbiome, microbial SCFAs productions, lactic acid and serum metabolites in broiler chickens . the results showed that supplementation with 200 mg/kg EM improved the immune response, increased cecum short-chain fatty acids (SCFAs) and lactate concentrations, and improved nutrient digestibility in broilers . moreover , EMIE altered the composition of cecal microbiome, increasing the relative abundance of beneficial bacteria (Candidatus_Soleaferrea and Lachbospiraceae_NC2004_group and Butyricioccus) and reducing that of harmful bacteria (UBA1819, Negativibacillus, and Eisenbergiella). I think that the manuscript is convenient with the scope of the journal. The paper could provide information of interest in this field.

But why authors used one level from the extract (200mg) and they should mention that in the manuscript ???

Author Response

Comments from Reviewer 1

The present study was conducted to investigate the main isopentenyl flavonols in EM and whether their synergistic interactions affect nutrient digestibility, the immune response, the cecum microbiome, microbial SCFAs productions, lactic acid and serum metabolites in broiler chickens . the results showed that supplementation with 200 mg/kg EM improved the immune response, increased cecum short-chain fatty acids (SCFAs) and lactate concentrations, and improved nutrient digestibility in broilers . moreover , EMIE altered the composition of cecal microbiome, increasing the relative abundance of beneficial bacteria (Candidatus_Soleaferrea and Lachbospiraceae_NC2004_group and Butyricioccus) and reducing that of harmful bacteria (UBA1819, Negativibacillus, and Eisenbergiella). I think that the manuscript is convenient with the scope of the journal. The paper could provide information of interest in this field.

But why authors used one level from the extract (200mg) and they should mention that in the manuscript ???

Response:We are thankful for the reviewer’s positive comments on our work. In our previous work, we conducted tests for the optimal additive amount of Epimedium extract and optimized it using growth performance as the main parameter. Results showed that 200 mg/kg of Epimedium extract had the best growth performance and other health improving indicators. This part has been published in “Journal of Animal Science and Biotechnology”, and we have cited this article as the reference (No.22) which counld be found  in lines 82-84, respectively.

Reviewer 2 Report

Dear author

Thanks for your work and nice presentation.

However, some comments should be considered before accepting the manuscript;

  1. The title could be changed without adding the verb revealed. For example: The effect of Epimedium isopentenyl flavonoids on the broiler gut health using microbiomic and metabolomic analyses.
  2. Though some abbreviations such as EM, SCFAs, UHPLC-Q-TOF/MS, Cr2O3, DB-FFAP, PCR, etc. have been mentioned” as full names” in the abstract section, they have not been mentioned in the text of the manuscript. So, please mention them at the first time as full words.
  3. What about the graphic abstract or figure in the page (2)? It has not been mentioned the text.
  4. The duration of the each treatment EM or tetracycline should be mentioned in the experimental design.
  5. The author supplemented the basal diet with 75 mg/kg chlortetracycline as a control group to compare it with EM treatment.  Are the antibiotic fed additives still added to the diets of birds? You should simulate the field conditions. Explain.
  6. Sample collection: Details regarding blood samples “timing, site of collection, etc.” as well for the caecum samples “timing or the bird’s age, number of sampling times, etc.) should be mentioned in details.
  7. Chicken “droppings” not manure, feces, or stool!.
  8. Immune markers: Interleukin-6 (IL-6), interleukin-10 (IL10), interleukin1β (IL-1β). One you mention the first abbreviation as full words, you should not repeat it. For example; Interleukin-6 (IL-6), IL10, and IL-1β. The same for immunoglobulins. However, ELISA should be mentioned at the first time as full words.
  9. More details about Shannon index of cecum microorganisms could be mentioned.
  10. Why the clinical parameters such as the performance ones “BWG, FCR, etc.” have not been measured to correlate the positive changes in the gut microbiome with the production status of the birds as in the field. The same for the immune response, why the antibody titers have not been mentioned for the most important viral diseases of poultry such as NDV, AIV, IB, etc.?
  11. What is the cost: benefit ratio of using EM product in the field of poultry industry? 

Best wishes

Author Response

Reviewer 2

Thanks for your work and nice presentation.

However, some comments should be considered before accepting the manuscript;

Response:Thank you for your support and valuable comments. Your comments are very helpful for improving this manuscript and providing novel insights to our research. We have carefully revised the manuscript based on your comments and suggestions.

Comment 1: The title could be changed without adding the verb revealed. For example: The effect of Epimedium isopentenyl flavonoids on the broiler gut health using microbiomic and metabolomic analyses.

Response: Thank you for your valuable suggestion We have changed the Title to " The effect of Epimedium isopentenyl flavonoids on the broiler gut health using microbiomic and metabolomic analyses " in the revised manuscript.

Comment 2: Though some abbreviations such as EM, SCFAs, UHPLC-Q-TOF/MS, Cr2O3, DB-FFAP, PCR, etc. have been mentioned” as full names” in the abstract section, they have not been mentioned in the text of the manuscript. So, please mention them at the first time as full words.

Response: Thanks for your comments. We have changed the abbreviation to the full name when it was mentioned at the first time in the text of the revised manuscript.

Comment 3: What about the graphic abstract or figure in the page (2)? It has not been mentioned the text.

Response: Thanks for your comments. The graphical abstract on page 2 is our summary of the article, which was placed at the beginning to make it easier for readers to understand the overall content of our research.

Comment 4: The duration of the each treatment EM or tetracycline should be mentioned in the experimental design.

Response: Thank you for your comments. The duration of each treatment EM or tetracycline has been added in the section of “Birds and experimental design”.

Comment 5: The author supplemented the basal diet with 75 mg/kg chlortetracycline as a control group to compare it with EM treatment.

Response 5: Thank you for the professional comments. Prior to the ban of growth-promoting antibiotics, in many countries around the world, chlortetracycline was commonly used as an antibiotic additive in livestock farming, with a recommended dose of 75 mg/kg. This was achieved by first pre-mixing of antibiotics, minerals, vitamins, amino acids and carriers to create a premix, and then the premix was used to produce a feed product with other feed ingredients. After the ban of antibiotics, plant extracts were chosen as potential alternative products to improve the health and performance of broiler chickens. Epimedium is one of such plant extracts. In order to evaluate its effects, we selected chlortetracycline as a positive control group in this study because of its ability to prevent diseases and enhance the growth of broiler chickens.

Comment 6: Sample collection: Details regarding blood samples “timing, site of collection, etc.” as well for the caecum samples “timing or the bird’s age, number of sampling times, etc.) should be mentioned in details.

Response 6: Thank you for your comments. The detailed information has been added in the section of “Sample and Collection” in the revised manuscript.

Comment 7: Chicken “droppings” not manure, feces, or stool!

Response 7: Thanks. We have changed “feces” and “stool” to “dropping” in the revised manuscript.

Comment 8: Immune markers: Interleukin-6 (IL-6), interleukin-10 (IL10), interleukin1β (IL-1β). One you mention the first abbreviation as full words, you should not repeat it. For example; Interleukin-6 (IL-6), IL10, and IL-1β. The same for immunoglobulins. However, ELISA should be mentioned at the first time as full words.

Response 8: Thanks for your comments. Thses have been modified in this sentence according to the reviewer’s suggestion.

Comment 9: More details about Shannon index of cecum microorganisms could be mentioned.

Response 9: Thank you for your comments. To make the article more concise, detailed information about the Shannon index of cecum microbiota had been added in the supplemental material. And it also was pointed out in the section “The 16S rRNA high-throughput sequencing”, as follows: Full details of the methods are reported in Supplementary Information 4.

Comment 10: Why the clinical parameters such as the performance ones “BWG, FCR, etc.” have not been measured to correlate the positive changes in the gut microbiome with the production status of the birds as in the field. The same for the immune response, why the antibody titers have not been mentioned for the most important viral diseases of poultry such as NDV, AIV, IB, etc.?

Response 10: Thanks for your professional comments. Our initial goal of this research was to systematically evaluate the effectiveness of EM in improving broiler health and growth performance, and to elucidate the mechanisms by which EM exerts its beneficial effects from metabolic and microbiological perspectives. This was a very large and complex project, and it was not feasible to present it in one article. Therefore, we divided our results into two parts: the first part focused on the optimization of the additive dosage of EM based on growth performance parameters, and it has been published in the Journal of Animal Science and Biotechnology; the second part, which is this manuscript, mainly used the results of the first part as a basis to explain the mechanisms of health improvement by Isopentenyl flavonol from metabolic and microbiomic viewpoints. Hence, we did not include growth performance parameters in this manuscript. Moreover, we aimed to provide valuable data on the potential use of EM as a nutraceutical. To ensure the relevance and comprehensiveness of our findings, we carefully selected common indicators of immune response, such as IgA, IgG, IL-10, IL-6, IL-1β, and TNF-α, rather than focusing solely on specific indicators of poultry immunity. We hope this explanation is satisfactory. Thank you again.

Comment 11: What is the cost: benefit ratio of using EM product in the field of poultry industry?

Response 11: Thanks, the cost of adding Epimedium as a feed additive increased by 4 USD for every 1000 kg of feed, which was slightly higher than that of chlortetracycline. However, it is unfortunate that antibiotics (including chlortetracycline) are not allowed to be added in feed as feed additives in many countries, such as China, United States, and Europe. In addition, Epimedium as one of the alternatives to antibiotics does not pose problems of environmental pollution, bacterial resistance, food safety, etc. This is very important and the value it brings is immeasurable. Moreover, with advances in cultivation techniques and science, the cost of plant extracts as feed additives is expected to decrease significantly. Thank you again.

Reviewer 3 Report

Firstly, I must apologise for what looks like a massive number of corrections. This is an excellent paper and my corrections are all textual, related to the correct wording of taxonomy, listing the methods used to more detail and some figure text corrections. The manuscript is very impressive and I congratulate the authors on a good job and on their attention to detail in this study. My comments are below.

Abstract:

  • “Epimedium, also known as barrenwort
  • Re Candidatus_Soleaferrea -  Please remove the underscore in all taxonomy. R packages replace space with an underscore because of space-delimited files so Lactobacillus agilis becomes Lactobacillus_agilis. This is R artefact, and it is fundamentally taxonomically wrong. Please correct this in the text and in all figures. In your case Candidatus_Soleaferrea should be Candidatus Soleaferrea, and everywhere else in the text. Taxonomy must be correct.
  • “cecum flora” - Please remove all instances of the word microflora and flora. If you google the definition of “flora” you will see that it is all forms of PLANT life. There are no microscopic plants in the gut. Calling microbiota or microbial population as "microflora" started in online journalism by non-academic authors and took root in academic manuscripts as a terrible misuse of academically accepted taxonomy. Please use microbiota instead of microflora everywhere in text

Introduction:

  • Introduction is well presented but quite short – 2 paragraphs only, one para on isoflavanoids and one on the plant. I assume that there is not much literature relevant to your findings that can set the reader to better understand your data?

Methods:

  • “15 birds per barn” barn is a house like structure, a poultry shed, I think you mean 15 birds per pen? That would be 8 pens with 15 birds per pen? That does not add up to total of 350 birds. Can you please be more specific?
  • So you have 3 groups – control, antibiotic and EM. This needs to be spelled out clearly
  • “Cecum contents were collected for microbiome analysis”. 16 S is not metagenomic sequencing, you are working with microbiota and not with the microbiome.  Please do not use the word microbiome wrongly for 16S data. 16S amplicon data gives us only taxa. Confusing 16S rRNA gene amplicon sequencing (microbiota) with metagenomics and microbiome is a very common issue. Microbiome covers everything living in the microbial community (microbiota/taxonomy), their collective genetic and functional potential (via metagenome), their products (metabolome) and environmental interactions, and it is a much more intricate model. A panel of international experts in the Microbiome Support Project (MSP) comprised of 40 leaders from all microbiome areas and more than a hundred invited international experts. The manuscript describing MSP panel recommendations (Berg et al., 2020) insists on clarifying differences between microbiota and microbiome, providing many currently accepted, somewhat opposing definitions of the microbiome; microbiome includes microbes (microbiota), their products, proteins, genomes, transcriptomes, metabolites and environment variables capable of capturing microbe-environment interactions (Berg et al., 2020). To get a view of the microbiome we need to do functional analysis via shotgun sequencing and a minimum, and if possible metabolomics. Unfortunately this term is used wrongly more than it is used correctly. Please correct throughout the text like “Microbiome analysis was performed” should be “Microbiota analysis was performed”
  • Continuing to previous point, when you are talking about 16S it is microbiota, but I can see you did metabolites and PiCrust for function so you must distinguish when it is OK to say microbiome, in your case not for 16S amplicon sequencing but it is ok when you are talking about overall analysis, I hope I was able to explain this concept…
  • “Cr2O3 content, moisture, total energy, crude protein, calcium, phosphorus, etc. in the feed and faeces were measured” – how were they measured, this should be thoroughly described in methods, the kits used, catalogue numbers and suppliers.
  • Please specify QIIME2 protocol, did you denoise? did you remove the chimaera? what was the quality score of your selected sequences? Did you use GreenGenes or Silva taxonomy? Please remember that GreenGenes is outdated with no updates for 10 years and does not have current taxonomy…
  • Which R package did you use for a correlation network to calculate correlations?

Results:

  • Data is very well presented, well done
  • Fonts in figure 5 must be made bigger and bolder, it is hard to read fig 5D even when zooming in
  • Figure 6 is great
  • Figure 7 should be zoomed in as much as the page allows
  • In the network remove underscores and the extra text. This is easy to edit in Cystoscape.  For example g_unclassified_Oscillospiraceae should be Un. Oscillospiraceae. In some cases, the IDs are much longer. Also genus and species must be italic, and other levels taxonomy as per journal rules
  • Piecrust is not described in methods, it just appears in the results

Discussion:

  • Very powerful discussion which is not a surprise because you have so much good data

Finally, I wish the authors all the best in their future research. This will be an excellent paper when it is corrected and I am sure I will cite it in my own work. Well done!

Author Response

Reviewer 3

Firstly, I must apologise for what looks like a massive number of corrections. This is an excellent paper and my corrections are all textual, related to the correct wording of taxonomy, listing the methods used to more detail and some figure text corrections. The manuscript is very impressive and I congratulate the authors on a good job and on their attention to detail in this study. My comments are below.

Response:Thanks for your professional and helpful comments. We have carefully revised the manuscript based on your comments and suggestions, and details could be found in the revised manuscript.

Comment 1: Abstract:“Epimedium, also known as barrenwort”

Response 1: Thanks for your suggestion. We have modified the description in "Abstract" in line 16.

Comment 2: Abstract: Re Candidatus_Soleaferrea - Please remove the underscore in all taxonomy. R packages replace space with an underscore because of space-delimited files so Lactobacillus agilis becomes Lactobacillus_agilis. This is R artefact, and it is fundamentally taxonomically wrong. Please correct this in the text and in all figures. In your case Candidatus_Soleaferrea should be Candidatus Soleaferrea, and everywhere else in the text. Taxonomy must be correct.

Response 2: Thanks for your comments. We have changed the error in the full text.

Comment 3: Abstract: “cecum flora” - Please remove all instances of the word microflora and flora. If you google the definition of “flora” you will see that it is all forms of PLANT life. There are no microscopic plants in the gut. Calling microbiota or microbial population as "microflora" started in online journalism by non-academic authors and took root in academic manuscripts as a terrible misuse of academically accepted taxonomy. Please use microbiota instead of microflora everywhere in text.

Response 3: Thanks for your comments. We have revised the presentation error in the full text.

Comment 4: Introduction: Introduction is well presented but quite short – 2 paragraphs only, one para on isoflavanoids and one on the plant. I assume that there is not much literature relevant to your findings that can set the reader to better understand your data?

Response 4: Yse, it is really that there are not enough related literatures could be found.  And in order to help readers to understand our work clearer, the content of  how isopentenyl flavonoids from other plants  affect host intestinal flora and plasma metabolism were added in the section of Introduction and  corresponding references are also supplemented. Thanks.

Comment 5: Methods:“15 birds per barn” barn is a house like structure, a poultry shed, I think you mean 15 birds per pen? That would be 8 pens with 15 birds per pen? That does not add up to total of 360 birds. Can you please be more specific?

Response 5: Thankwfor your comments. We have revised the description of the test method to make the method clearer and more explicit in the line 117.

Our experiment was divided into 3 groups of 8 replicates each, with 15 chickens in each replicate. Therefore 3 groups  8 replicates  15 birds/per replicate = 360 birds.

Comment 6: Methods: “Cecum contents were collected for microbiome analysis”. 16 S is not metagenomic sequencing, you are working with microbiota and not with the microbiome.  Please do not use the word microbiome wrongly for 16S data. 16S amplicon data gives us only taxa. Confusing 16S rRNA gene amplicon sequencing (microbiota) with metagenomics and microbiome is a very common issue. Microbiome covers everything living in the microbial community (microbiota/taxonomy), their collective genetic and functional potential (via metagenome), their products (metabolome) and environmental interactions, and it is a much more intricate model. A panel of international experts in the Microbiome Support Project (MSP) comprised of 40 leaders from all microbiome areas and more than a hundred invited international experts. The manuscript describing MSP panel recommendations (Berg et al., 2020) insists on clarifying differences between microbiota and microbiome, providing many currently accepted, somewhat opposing definitions of the microbiome; microbiome includes microbes (microbiota), their products, proteins, genomes, transcriptomes, metabolites and environment variables capable of capturing microbe-environment interactions (Berg et al., 2020). To get a view of the microbiome we need to do functional analysis via shotgun sequencing and a minimum, and if possible metabolomics. Unfortunately, this term is used wrongly more than it is used correctly. Please correct throughout the text like “Microbiome analysis was performed” should be “Microbiota analysis was performed”

Response 6: Thanks for your professional suggestion. We have changed the description error and checked the full text.

Comment 7: Methods: Continuing to previous point, when you are talking about 16S it is microbiota, but I can see you did metabolites and PiCrust for function so you must distinguish when it is OK to say microbiome, in your case not for 16S amplicon sequencing but it is ok when you are talking about overall analysis, I hope I was able to explain this concept…

Response 7: Thanks for your comments. You gave very detailed and clear instructions. 16S amplicon data gives us only taxa, while microbiota has a broader meaning. We have reworked the wording specification to ensure that there are no academic errors.

Comment 8: Methods: “Cr2O3 content, moisture, total energy, crude protein, calcium, phosphorus, etc. in the feed and faeces were measured” – how were they measured, this should be thoroughly described in methods, the kits used, catalogue numbers and suppliers.

Response 8: Thanks for your comments. We have supplemented the method with a description of the assay and corresponding reagents in lines 137-156 and lines 97-99.

Comment 9: Methods: Please specify QIIME2 protocol, did you denoise? did you remove the chimaera? what was the quality score of your selected sequences? Did you use GreenGenes or Silva taxonomy? Please remember that GreenGenes is outdated with no updates for 10 years and does not have current taxonomy…
Methods: Picrust2 is not described in methods, it just appears in the results.

Response 9: Thanks for your comments. Based on the default parameters, we performed quality control and filtering on the raw sequences and used the DADA2 plugin in Qiime2 to remove noise and generate amplicon sequence variants (ASVs). The quality score of our selected sequences was 95%. We used Silva as the reference database for taxonomic classification of the ASVs. We added the description of pircrust2 in "Methods". More detailed information can be found in Supplementary Information 4.

Comment 10: Methods: Which R package did you use for a correlation network to calculate correlations?

Response 10: Thanks for your comments. The software we used to calculate the correlation matrix is R software. During the calculation, we used the calculation function that comes with the R software and did not install an additional R package with the following code.

otu<-read.table('data.csv',sep=',',row.names = 1,header = T)

data<- t(otu)

otu.cor<- cor(data, method = "spearman", use = "complete.obs")

otu.p<- cor(data, method = c("spearman"))

otu.corvalue<- otu.cor[1:76,77:135]

otu.pvalue<- otu.p[1:76,77:135]

Comment 11: Results: Data is very well presented, well done

Fonts in figure 5 must be made bigger and bolder, it is hard to read fig 5D even when zooming in

Response 11: Thanks for your comments. We have adjusted the font size in all figures to ensure that the font is at least 6pt when the journal is 190mm wide, in line with journal standards.

12: Results: Figure 6 is great. Figure 7 should be zoomed in as much as the page allows

Response 12: Thanks for your comments. We have enlarged the size of Figure 7 as much as possible. In addition, we can read Supplementary Material Table 2 to help us understand Figure 7. Supplementary Material Table 2 can help us see exactly how much variation there is between the groups.

Comment 13: In the network remove underscores and the extra text. This is easy to edit in Cystoscape.  For example g_unclassified_Oscillospiraceae should be Un. Oscillospiraceae. In some cases, the IDs are much longer. Also genus and species must be italic, and other levels taxonomy as per journal rules

Response 13: Thanks for your comments. We have modified the genera and species according to the taxonomic format and italicized them in the network and full text.

Comment 14: Discussion: Very powerful discussion which is not a surprise because you have so much good data

 Finally, I wish the authors all the best in their future research. This will be an excellent paper when it is corrected and I am sure I will cite it in my own work. Well done!

Response 14: Thank you very much for the recognition of our work. Your comments and suggestions are very professional, and we learned a lot from it. We have carefully revised the manuscript according to your suggestions in order to make our description more academic and standard. Thanks again!

Round 2

Reviewer 2 Report

Dear author(s).

Thanks for a good response to the comments.

Best wishes 

Author Response

Thank you for your support and valuable comments. Your comments are very helpful for improving this manuscript and providing novel insights to our research. Thank you again for your careful review!

Reviewer 3 Report

The authors have corrected manuscript very thoroughly, the only thing that needs to be updated is the details of 16S analysis, saying it was done in QIIME2 is not enough. Please specify denoising, taxonomy database and other relevant analysis methodologies. 

After this one sentence is amended the manuscript is good to publish.

Author Response

Response:We have updated the details of the 16S analysis, including the specific work of QIIME2, and described denoising, taxonomy database and other relevant analysis methodologies. The modifications are shown in Supplementary Information 4, as follows.

Supp. Information 4: Details of the 16S rRNA high-throughput sequencing.

Total genomic DNA from the cecum samples was extracted by QIAamp DNA Stool Mini Kit (Qiagen, Hilden, Germany). DNA concentration and integrality were detected by NanoDrop Spectrophotometer (Thermo Scientific, Wilmington, DE, USA) and agarose gel electrophoresis, respectively. DNA concentration of each cecal content was diluted to 10 ng/μL using double-distilled water. The V3–V4 region of 16S rDNA was amplified using the following specific primers (338F: 5´-ACTCCTACGGGAGGCAGCAG-3´; 806R: 5´-GGACTACHVGGGTWTCTAAT-3´). Purified amplicons were pooled in equal amounts and paired-end sequenced (2×250 bp) on an Illumina MiSeq platform at Majorbio Bio-Pharm Technology Co. Ltd. (Shanghai, China).

The raw data was uploaded to the NCBI Sequence Read Archive databas (Accession Number: PRJNA904521). QIIME2 (version 2021.8 or higher, http://qiime2.org) was used for quality control and analysis of sequence reads.Raw fastq files were demultiplexed using q2-demux and quality filtered and dereplicated with q2-dada2. Sequences with average Phred scores below 25 were removed. The amplified subsequence variants (ASVs) were classified using the q2-feature classifier classify-sklearn naïve Bayes classifier on the SILVA database version 132 (https://www.arb-silva.de/download/archive/qiime/). MAFFT was used for sequence comparison of multiple ASVs by q2-alignment, and phylogeny was constructed by q2-phylogeny using fasttree2. To calculate α- and β-diversity, data were diluted to the lowest number of sequences possible for each sample. α-diversity was measured by the Shannon index. differences between α-diversity indices were tested by the Kruskal-Wallis test (QIIME 2). To estimate the similarity of microbial community structure (β-diversity) between groups, a principal coordinate analysis (PCoA) based on a weighted UniFrac distance matrix (QIIME 2) was performed. Partial least squares discriminant analysis (PLS-DA) was conducted to compare the bacterial community structures across all samples. Moreover, the significance of differentiation of microbial structure among groups was statistically tested by analysis of similarity. The linear discriminant analysis (LDA) was coupled with effect size measurements (LEf-Se) to distinguish the bacteria between all the treatments, the LDA score was set at two. To conduct functional prediction analysis of the 16S data, data analysis was performed through the Phylogenetic Investigation of Communities by Reconstruction of Unobserved States (PICRUSt)2 pipeline.